# Outcomes of Extracorporeal Membrane Oxygenation for Acute Respiratory Distress Syndrome in COVID-19 Patients: A Propensity-Matched Analysis

**DOI:** 10.3390/jcm10122547

**Published:** 2021-06-09

**Authors:** Teresa Autschbach, Nima Hatam, Koray Durak, Oliver Grottke, Michael Dreher, Katharina Nubbemeyer, Rolf Rossaint, Gernot Marx, Nikolaus Marx, Jan Spillner, Rashad Zayat, Sebastian Kalverkamp, Alex Kersten

**Affiliations:** 1Department of Thoracic and Cardiovascular Surgery, RWTH University Hospital Aachen, Medical Faculty, RWTH Aachen University, Pauwelsstr. 30, 52074 Aachen, Germany; rautschbach@ukaachen.de (T.A.); nhatam@ukaachen.de (N.H.); koray.durak@outlook.de (K.D.); knubbemeyer@ukaachen.de (K.N.); jspillner@ukaachen.de (J.S.); skalverkamp@ukaachen.de (S.K.); 2Department of Anesthesiology, RWTH University Hospital Aachen, Medical Faculty, RWTH Aachen University, Pauwelsstr. 30, 52074 Aachen, Germany; ogrottke@ukaachen.de (O.G.); rrossaint@ukaachen.de (R.R.); 3Department of Pneumology and Intensive Care Medicine, RWTH University Hospital Aachen, Medical Faculty, RWTH Aachen University, Pauwelsstr. 30, 52074 Aachen, Germany; mdreher@ukaachen.de; 4Department of Intensive Care and Intermediate Care Medicine, RWTH University Hospital Aachen, Medical Faculty, RWTH Aachen University, Pauwelsstr. 30, 52074 Aachen, Germany; gmarx@ukaachen.de; 5Department of Cardiology, Angiology and Intensive Care, RWTH University Hospital Aachen, Medical Faculty, RWTH Aachen University, Pauwelsstr. 30, 52074 Aachen, Germany; nmarx@ukaachen.de (N.M.); akersten@ukaachen.de (A.K.)

**Keywords:** SARS-CoV-2, acute respiratory distress syndrome, extracorporeal membrane oxygenation, COVID-19, thromboembolic events, coagulation

## Abstract

It remains unclear to what extent the outcomes and complications of extracorporeal membrane oxygenation (ECMO) therapy in COVID-19 patients with acute respiratory distress syndrome (ARDS) differ from non-COVID-19 ARDS patients. In an observational, propensity-matched study, outcomes after ECMO support were compared between 19 COVID-19 patients suffering from ARDS (COVID group) and 34 matched non-COVID-19 ARDS patients (NCOVID group) from our historical cohort. A 1:2 propensity matching was performed based on respiratory ECMO survival prediction (RESP) score, age, gender, bilirubin, and creatinine levels. Patients’ characteristics, laboratory parameters, adverse events, and 90-day survival were analyzed. Patients’ characteristics in COVID and NCOVID groups were similar. Before ECMO initiation, fibrinogen levels were significantly higher in the COVID group (median: 493 vs. 364 mg/dL, *p <* 0.001). Median ECMO support duration was similar (16 vs. 13 days, *p =* 0.714, respectively). During ECMO therapy, patients in the COVID group developed significantly more thromboembolic events (TEE) than did those in the NCOVID group (42% vs. 12%, *p =* 0.031), which were mainly pulmonary artery embolism (PAE) (26% vs. 0%, *p =* 0.008). The rate of major bleeding events (42% vs. 62%, *p =* 0.263) was similar. Fibrinogen decreased significantly more in the COVID group than in the NCOVID group (*p <* 0.001), whereas D-dimer increased in the COVID group (*p =* 0.011). Additionally, 90-day mortality did not differ (47% vs. 74%; *p =* 0.064) between COVID and NCOVID groups. Compared with that in non-COVID-19 ARDS patients, ECMO support in COVID-19 patients was associated with comparable in-hospital mortality and similar bleeding rates but a higher incidence of TEE, especially PAE. In contrast, coagulation parameters differed between COVID and NCOVID patients.

## 1. Introduction

As of 4 February 2021, severe acute respiratory syndrome coronavirus 2 (SARS-CoV-2) has infected >105 million people and caused more than two million deaths worldwide [1]. In cases with refractory hypoxemic respiratory failure and insufficient improvement despite mechanical ventilation (MV), extracorporeal membrane oxygenation (ECMO) support is a life-rescuing treatment per the World Health Organization [2]. ECMO therapy has gained acceptance as beneficial rescue therapy in acute respiratory distress syndrome (ARDS) patients [3,4]. During the 2009 influenza A (H1A1) pandemic, ARDS patients benefited from ECMO support [5]. Data on ECMO therapy of critically ill COVID-19 patients are scarce; hence, the effects of ECMO support in COVID-19 ARDS patients are uncertain. ECMO support presents an increased risk of bleeding and thromboembolic events (BE and TEE, respectively) [6]. SARS-CoV-2 also poses a high risk for TEE since 31% of critically ill COVID-19 patients developed arterial and venous thromboembolism [7,8]. Therefore, treating COVID-19 infection with ECMO treatment may negatively synergize the effect of increased TEE and BE. Data regarding outcome and complication, including TEE and BE, incidences during ECMO support for ARDS in COVID-19 patients remain scarce. A need is foreseen for further analysis of ECMO support of COVID-19 patients and the resulted changes of the coagulation factors. Consequently, we conducted this study to analyze and compare the outcomes and adverse events of ECMO therapy between COVID-19 ARDS and non-COVID-19 ARDS patients.

## 2. Materials and Methods

In an observational, propensity-matched cohort study, data from critically ill COVID-19 patients suffering from ARDS who were treated in our institution with venovenous (VV) ECMO between March 2020 to May 2020 were prospectively collected (*n =* 19). We reviewed our institution database for all ARDS patients (age >18 years old) who received ECMO therapy between January 2015 and January 2020. According to the Berlin Definition of the European Society of Intensive Care Medicine from 2011 [9], severe cases of ARDS were identified. The criteria for ECMO initiation in COVID-19 patients were (a) patients commonly accepted ECMO indications, as suggested by the Extracorporeal Life Support Organization (ELSO) [10] and (b) all other treatments options were exhausted, namely, lung-protective MV (MV), prone positioning, neuromuscular blockade, and inhaled nitric oxide rescue therapy (iNO).

The database included 105 ARDS patients. The ethics committee of RWTH University Hospital (EK 093/20) approved the study. The requirement for informed consent was waived by the ethics commission board because of the urgent need for COVID-19 data.

### 2.1. ECMO Settings

Our ECMO administration approaches, including configuration and the applied techniques, were recently published [11]. We used iLA activve^®^ Pumpe (XENIOS, Heilbronn, Germany) and Cardiohelp HLS Systems Version 7.0 (Maquet Cardiopulmonary GmbH, Rastatt, Germany) ECMO pumps. Percutaneous cannulation with the Seldinger technique was our preferred technique for VV ECMO. Depending on the desired flow rate and where possible, bicaval cannulation with a double lung cannula (27 to 31 Fr) was performed preferentially to two-site cannulation (femoral–jugular or femoral–femoral), with 19 to 25 Fr cannulation. The decision of whether to perform double-lumen cannulation on a single-site or two-place cannulation depends on many factors. In short, a 25 Fr vein cannula as a drainage cannula is generally necessary for a patient with high BSA (2.2–2.5 m^2^), and a 17–19 Fr vein cannula for venous return is appropriate to achieve sufficient flow with adequate carbon dioxide clearance and oxygenation.

### 2.2. Anticoagulation

Hemostasis parameters were measured daily and included the activated partial thromboplastin time (aPTT), international normalized ratio (INR), platelet count, fibrinogen, antithrombin III, D-dimer, and activated clotting time (ACT). ACT was measured three times per day as a control for coagulation, and factor XIII was measured three times per week. We used unfractionated heparin (UFH) for anticoagulation management of ECMO patients with no contraindications. In VV ECMO patients, we aimed for aPTT 40–50 s or 160–180 s ACT. If necessary, UFH was reduced or paused. Other target values were <1.4 INR, >50 G/L platelet count, and >150 mg/dL fibrinogen. We adjusted target values using fresh frozen plasma or packed red blood cells (PRBCs) in the case of bleeding. We reduced the ACT target to <160 s, normalized INR, and raised the platelet count to ≥80 G/L and fibrinogen to >200 mg/dL.

Many recent studies have shown that COVID-19 patients have an increased risk for thromboembolic events [6,7,8]. We adapted our anticoagulation strategy and aimed at higher ACT and aPTT targets (aPTT 50–60 s and ACT 180 s). This was also recommended by ELSO guidelines [10]. PRBC transfusion was guided according to two parameters: targeted hemoglobin levels of >9 g/dL [12] and sufficient oxygen delivery alongside ultraprotective ventilation. The ratio of oxygen delivery (Do_2_) to oxygen consumption (Vo_2_) was monitored to ensure ≥3:1, ideally >4:1. PRBCs were transfused if Do_2_:Vo_2_ ratios were poor, even when hemoglobin was ~10 g/dL.

Patient demographics, laboratory data, MV parameters, ECMO settings, clinical course, adverse events, and outcomes were compiled. Laboratory tests included complete blood count, chemistry panel, and hemostasis-related parameters (hemoglobin [Hb], leucocytes, platelet count, procalcitonin [PCT], aPTT, INR, fibrinogen, D-dimer, anti-thrombin III [ATIII], free plasma hemoglobin [fpHb], lactate dehydrogenase [LDH], and aspartate aminotransferase [AST]). We analyzed laboratory data from before ECMO initiation, 24 h after ECMO implantation, and on the final day of ECMO support before de-cannulation.

Our primary endpoint was in-hospital mortality, and the secondary endpoints were major adverse events occurring within 90 days after ECMO initiation. Major bleeding events were defined according to the International Society on Thrombosis and Hemostasis (ISTH) [13] as acute decline (<1 h) of Hb levels of >1.24 mmol/L or the need for >2 units PRBC transfusion. If TEE were suspected based on clinical symptoms and physical examination, ultrasound, CT scans, and if necessary, MRI were conducted to secure the diagnosis of TEE. To detect SARS-CoV-2, throat swab specimens, tracheal secretions, or bronchoscopic alveolar lavage were obtained upon hospital admission. Real-time reverse transcription–polymerase chain reaction further confirmed COVID-19 infections.

### 2.3. Statistical Analysis

Categorical variables are presented as absolute numbers and percentages (%). Continuous variables were tested for normal distribution with the Kolmogorov–Smirnov test and were presented as median and interquartile range (IQR) values.

Despite the small COVID-19 cohort, we minimized confounding because of differences in patients’ baseline characteristics through propensity matching using the propensity score of the respiratory extracorporeal membrane oxygenation survival prediction (RESP) score, age, gender, bilirubin, and creatinine values pre-ECMO. From 20 ARDS COVID-19 patients, one patient had VA-ECMO and was excluded from the analysis. In the non-COVID ARDS group, 38 (36%) patients had VA-ECMO and were not included in the matching. We used a 1:2 matching approach with a maximum score radius of 0.05 to match between 19 ARDS COVID-19 patients (COVID group) and 67 non-COVID ARDS patients (NCOVID group). Four patients from the COVID group were able to be matched with one patient from the NCOVID group. After matching, we had 19 patients in the COVID group and 34 patients in the NCOVID group. The balance between the matched group was assessed with the standardized absolute bias difference (Appendix A). This matching procedure enabled higher precision at little cost of bias [14]. Comparisons between unmatched groups were performed with two-tailed Student’s *t*-tests for normally distributed continuous variables and with the Mann–Whitney U test for nonnormally distributed continuous variables. Categorical variables were analyzed with a chi-square test or, if appropriate, Fisher’s exact test.

In the matched cohort, univariate analyses were conducted using the Wilcoxon signed-rank test for continuous variables and McNemar’s test for categorical variables. Analyses of laboratory parameters at three-time points were conducted using Friedman’s nonparametric test with Dunn’s correction for repeated measurement. Adjusted *p* values for multiple corrections are presented. Crude survival by COVID-19 status was assessed with Kaplan–Meier analyses weighted for the propensity scores, and the test of equality of survival was carried out through Fleming–Harrington test; parametric survival regression adjusted for the propensity score was performed to estimate the hazard ratio (HR) and to correspond 95% confidence interval (CI). All statistical comparisons were two-sided, and a *p* of <0.05 was significant.

Propensity score calculation and case matching were performed with SAS software 9.4 (SAS Institute Inc., Cary, NC, USA). Statistical analysis was conducted using SPSS Version 26 (IBM Corp., Armonk, NY, USA). Kaplan–Meier survival estimates and parametric survival regression, including visualization, were obtained using open source software Jamovi version 1.2.22.0. Time courses were displayed with GraphPad Prism version 8.0 (GraphPad Software, San Diego, CA, USA).

## 3. Results

Table 1 presents the patient characteristics and laboratory results for 53 patients (19 COVID matched to 34 NCOVID). Median ages were 57 (IQR 50–62) and 55 (IQR 50–62) years old in the COVID and NCOVID groups, respectively (*p =* 0.914). The median length of hospitalization before ECMO implantation was six days (IQR 4–16) in the COVID group and one day (IQR 0–8) in the NCOVID group (*p =* 0.695). We reported two significant differences in patient characteristics. Nicotine abuse was significantly less frequent in the COVID group versus the NCOVID group, with 21% versus 53% (*p =* 0.008). COVID-19 patients had significantly more arterial hypertension in their medical history (74% vs. 41%, *p =* 0.013). All Patients’ characteristics before matching are presented in Appendix A.

### 3.1. Pre-ECMO Laboratory Parameters

Before ECMO initiation, COVID patients had significantly higher fibrinogen levels than NCOVID patients (493 [IQR 545–704] vs. 364 [IQR 276–490] mg/dL, *p <* 0.001). Before ECMO implantation, D-dimer levels were similar in the COVID and NCOVID groups (4345 [IQR 1810–11,806] vs. 2400 [IQR 1782–7030] ng/mL, *p =* 0.253), and the median LDH was 403 (IQR 312–578) in the COVID group and 466 (IQR 318–644) U/L in the NCOVID group (*p =* 0.589). PaCO_2_ values were >75 mmHg in six patients (32%) of the COVID group and 14 patients (41%) of the NCOVID group (*p =* 0.424). All other laboratory parameters were similar (Table 1).

### 3.2. Time Course of Hemostasis Parameters

Figure 1 presents the time course of hemostasis parameters. We reported comparable Hb levels before ECMO implantation (median: 9.8 vs. 9.6 g/dL, *p <* 0.001) and before ECMO explantation (median: 9.7 vs. 9.6 g/dL, *p <* 0.001) in the COVID group versus NCOVID group.

Platelet counts were significantly higher in the COVID group during ECMO therapy at 24 h after ECMO initiation (*p =* 0.05) and on the last day of ECMO support (*p =* 0.02) (Figure 1 and Appendix A). Platelet levels decreased significantly during ECMO support, compared with baseline values in the COVID group (median: 222/nL vs. 163/nL, *p <* 0.01) and in the NCOVID group (median: 131/nL to 86/nL, *p <* 0.01). D-dimer values were significantly higher in the COVID group than in the NCOVID group after ECMO support (*p =* 0.011). Fibrinogen levels significantly decreased in the COVID group after ECMO support, compared with baseline (median: 493 vs. 268 mg/dL, *p <* 0.001), whereas fibrinogen levels remained stable in the NCOVID group (Figure 1 and Appendix A).

There were no significant differences in infection parameters (procalcitonin and leucocytes), aPTT, or INR values between the two groups (Figure 1 and Appendix A). The time-course of aPTT during the first seven days on ECMO is presented as a Appendix A, showing that aPTT did not differ significantly between the two groups. Leucocyte levels were comparable before ECMO implantation (median: 13/nL vs. 11/nL, *p <* 0.001) and before ECMO explantation (median: 13/nL vs. 11.4/nL, *p <* 0.001) in COVID group versus NCOVID group. Plasma-free hemoglobin (pfHb) values increased after ECMO support, compared with pre-ECMO initiation in the COVID group (median: 36–80 mg/dL, *p =* 0.006) and NCOVID group (median: 48–120 mg/dL, *p =* 0.002). pfHb values were lower in the COVID group than in the NCOVID group 24 h after ECMO initiation (median: 45 vs. 68 mg/dL, *p =* 0.041).

### 3.3. Outcomes and Clinical Course

Table 2 presents all outcomes and complications. During a 90-day follow-up, mortality was 47% in the COVID group and 74% in the NCOVID group (Fleming–Harrington test of equality of survival, *p =* 0.153). Figure 2 presents the Kaplan–Meier survival curves weighted for propensity score.

The adjusted parametric survival regression patients in the COVID group had a hazard ratio of 3.1, *p =* 0.056 [95%-confidence interval: 0.99–9.96].

The median ECMO duration was 16 days (IQR 11–23) in the COVID group and 13 days (IQR 5–25) in the NCOVID group (*p =* 0.714). The length of hospitalization was similar at 29 days (IQR 14–55) versus 33 days (IQR 11–60), *p =* 0.563. Approximately 50% of our patients with thrombotic events were survivors. ECMO-supported patients in the COVID group had significantly higher TEE incidence than in the NCOVID group (42% vs. 9%, *p =* 0.031). PAE incidences were more often in the COVID-19 group (26% vs. 0%, *p =* 0.008). The incidence of major bleeding events was 42% in the COVID group and 62% in the NCOVID group (*p =* 0.263). The most frequent location was mucosal (airway) in the COVID group (16%) and cannulation side in the NCOVID group (26%). Furthermore, acute kidney failure occurred in 68% of the COVID group and 50% of the NCOVID group (*p =* 0.093), hemorrhagic shock in 0% of the COVID group and 12% of the NCOVID group (*p =* 0.125), and severe thrombocytopenia in 2% of the COVID group and 32% of the NCOVID group (*p =* 0.057).

## 4. Discussion

Despite the increased incidence of TEE in COVID-19 ARDS patients treated with ECMO, the 90-day mortality rate was not higher. On the contrary, we found a trend towards better survival in COVID-19 patients. Early studies reported high mortality rates in COVID-19 patients receiving ECMO therapy, which raised doubts about its utilization [15]. An international cohort study including 1,035 ECMO-supported COVID-19 patients showed 38% mortality [16]. In our study, the mortality of ECMO-supported COVID-19 was 47%. However, these rates are consistent with mortality rates of ECMO-supported non-COVID-19 patients [3,4,5].

Mortality in the non-COVID group was high (74%), despite a RESP score with a predicted survival of 33%–57%. Mortality rates may be associated with baseline characteristics (Table 1). First, smoking as a risk factor was not included in the RESP calculation, but the proportion of nicotine users in the non-COVID-19 group was 51%. Six patients from the NCOVID group (18%) were on immunosuppressive medication (two were leukemia patients), and all of them died during ECMO therapy.

### 4.1. Hemostasis Parameters and Thromboembolic Events

The risk for TEE in ARDS patients due to SARS-CoV-2 infection increased, compared with non-COVID ARDS patients [7,8,17]. Autopsies of COVID-19 patients in recent studies identified high incidences of TEE [17,18]. Our study results are similar to those of Helms et al., who reported significantly higher TEE and PAE rates (11.7 vs. 2.1%, *p <* 0.008) in COVID-19 ARDS patients, compared with those in non-COVID ARDS patients [8]. We identified differences in hemostasis parameters. Before ECMO initiation, fibrinogen levels were significantly higher in the COVID group than in the NCOVID group, whereas D-dimer values and platelet counts were similar. Compared with the NCOVID patients, D-dimer and platelet count increased significantly in the COVID group during ECMO support, whereas fibrinogen levels in the COVID group reached significantly lower levels at the end of ECMO support. Other studies reported increased fibrinogen values in COVID-19 patients when compared with those in non-COVID-19 patients [19,20]. The high incidence of TEE in COVID-19 patients is multifactorial and might be explained by systemic inflammatory reaction syndrome accompanied by endothelial injury, which is caused by the attachment of the virus to endothelial cells and viral replication, which play a crucial role in the high coagulation state [21]. This can lead to prothrombic endothelial dysfunction, which is supported by platelet activation and other factors. TEE occurred despite anticoagulation therapy with UFH. Similar to the findings of Zhou et al. and Helms et al., fibrinogen levels significantly increased in our COVID-19 cohort, compared with those in non-COVID-19 patients, which indicates that a systemic inflammatory reaction may have activated coagulation. This also explains the increased D-dimer levels during ECMO support in the COVID group, which reached significantly higher values than in our non-COVID-19 patients at the end of ECMO treatment. Routine coagulation markers, such as platelet count, aPTT, ATIII, and thrombin, cannot detect a procoagulant state [22]. Zou et al. evaluated the correlation between coagulation parameters and disease severity in 240 patients from Shanghai, China [23]. They similarly reported significantly higher fibrinogen and D-dimer levels in critically ill patients. We hypothesize that COVID-19 patients have a strong correlation between disease severity and coagulation dysfunction.

We found a significantly prolonged aPTT (24 h after ECMO implantation) in the NCOVID group, and the platelet count decreased in both groups (24 h after ECMO implantation and pre-ECMO explantation). We aimed to keep the platelet count >50/nL and aPTT levels >50 s. aPTT levels in the NCOVID group were higher but within the recommended range, and the difference was not clinically relevant. Overall, 11 patients (32%) in the NCOVID group developed severe thrombocytopenia during ECMO therapy. A recent systematic review showed that a decreased platelet count during ECMO therapy is common because of foreign surface contact, high shear stress, etc. [24]. The increased prevalence in our study could be influenced by two patients (6%) with leukemia and the usage of antibiotics because of bacterial infections.

### 4.2. Perspectives

Our results suggest early, more frequent screening for PAE in COVID-19 patients during ECMO support and before ECMO during MV are necessary. COVID-19 patients did not show an increased risk for bleeding. Therefore, anticoagulation regimes with higher aPTT and ACT targets might be indicated in these patients. There is an urgent need for further prospective studies with larger sample sizes to analyze the hemostatic profile and differences caused by SARS-CoV-2. Possible risk factors leading to ECMO treatment failure in COVID-19 patients should be identified so that preventive strategies, including anticoagulation regimes and prediction models to identify at-risk patients, can be developed.

### 4.3. Limitations

Although our results provide information on coagulation and hemostatic changes in COVID-19 patients, our study was limited by a small sample size, especially for the COVID-19 cohort. However, we reached a sample size of 53 by using one-to-two propensity matching. Despite accounting for baseline bias with propensity matching, we could not use all known risk factors in our propensity scores. Therefore, some degree of bias in patients’ characteristics remains evident between the two groups. A small number of COVID-19 patients is reasonable because the pandemic newly erupted in the year of the study, and therefore, numerous intervention strategies were explored to treat the novel virus.

In contrast, ECMO therapy was used only at specialized centers for select critically ill patients. The retrospective analysis of our historical ARDS groups has the risk of sample selection bias. Although we used the RESP score and gender in the propensity matching, only a few preexisting diseases differed, which could have influenced the outcome. Considering the observational nature of this study, we were unable to perform an in-depth analysis of the pathophysiological pathways leading to increased TEE incidence in COVID-19 patients.

## 5. Conclusions

ECMO offers a rescue treatment for critically ill COVID-19 patients suffering from refractory respiratory failure. ECMO support in COVID-19 patients is associated with comparable in-hospital mortality and similar bleeding rates but a higher incidence of TEE, especially PAE, and significant differences in coagulation markers, compared with that in non-COVID-19 ARDS patients. Further investigations are required to characterize the coagulation profile of COVID-19 patients to provide an appropriate coagulation strategy.

## Figures and Tables

**Figure 1 jcm-10-02547-f001:**
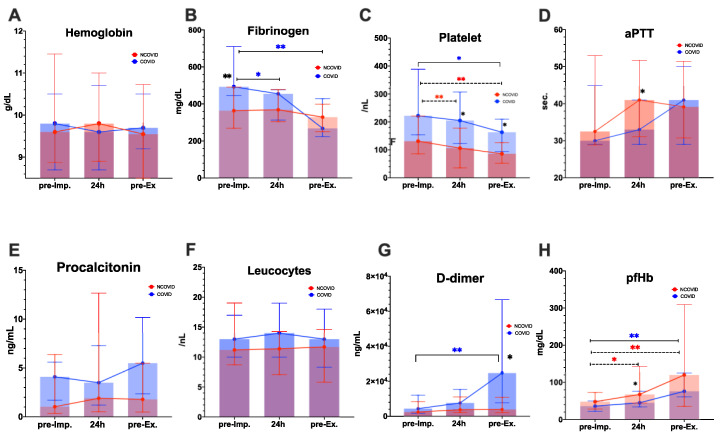
Time course of laboratory parameters. * *p* values < 0.05, ** *p* < 0.01 are significant and tagged with asterisks. Black asterisks indicate significance between the COVID and NCOVID groups. Colored asterisks show significance between two-time points. COVID group: *n =* 19; NCOVID group: *n =* 34. Abbreviations: aPTT—partial thromboplastin time; pfHb—plasma-free hemoglobin; pre-imp—pre-ECMO implantation; pre-Ex—pre-ECMO explantation.

**Figure 2 jcm-10-02547-f002:**
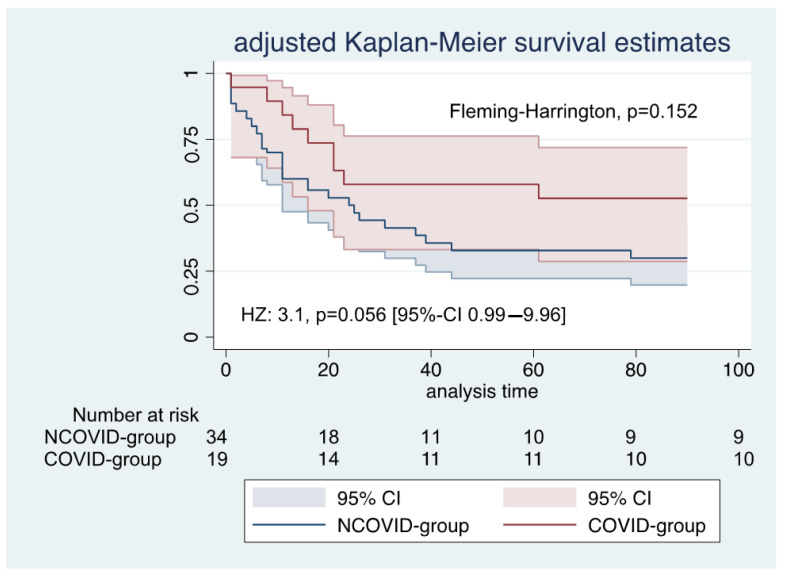
Kaplan–Meier survival estimates during the first 90 days after ECMO initiation: HZ—hazard ratio; 95% CI—95% confidence interval.

**Table 1 jcm-10-02547-t001:** Baseline characteristics.

	COVID (*n =* 19)	NCOVID (*n =* 34)	*p* Value
Age (y)	57 (50–62)	55 (50–62)	0.914
Female gender, *n* (%)	6 (32)	11 (32)	1.000
Weight (kg)	90 (80–100)	80.15 (70–100)	0.330
Height (cm)	176 (170–185)	172 (165–181)	0.177
BMI (kg/m^2^)	28.2 (24.7–31.1)	26.6 (25–33.4)	0.817
Pre-ECMO LOS in-hospital (d)	6 (4–16)	1 (0–8)	0.695
Coronary artery disease, *n* (%)	1 (5)	6 (18)	0.125
Prior myocardial infarction, *n* (%)	1 (5)	2 (6)	1.000
Arterial hypertension, *n* (%)	14 (74)	14 (41)	0.013 *
COPD, *n* (%)	3 (16)	10 (29)	0.267
Diabetes mellitus type 2, *n* (%)	7 (37)	5 (15)	0.096
Pulmonary hypertension, *n* (%)	2 (11)	0 (0)	0.250
Chronic kidney disease, *n* (%) †	2 (11)	3 (9)	1.000
Leukemia, *n* (%)	0 (0)	2 (6)	0.500
Nicotine use, *n* (%)	4 (21)	18 (53)	0.008 *
Immunosuppressive medication, *n* (%)	1 (5)	6 (18)	0.289
History of malignancy, *n* (%)	1 (5)	2 (6)	1.000
Blood Gas and Laboratory Tests			
pO_2_ (mmHg)	68 (54–72)	71 (51–82)	1.000
pCO_2_ (mmHg)	41 (35–51)	66 (48–79)	0.000 *
Lactate (mmol/L)	2.4 (1.1–4.6)	1.9 (1.3–3.0)	0.654
pH	7.4 (7.36–7.47)	7.3 (7.2–7.4)	0.003 *
HCO_3_^−^	25.6 (20–31)	27 (24.7–34.4)	0.108
FiO_2_	60 (50–80)	100 (55–100)	0.056
NO before ECMO (%)	10 (30)	14 (74)	0.001 *
NMB before ECMO (%)	33 (97)	11 (57)	0.04 *
FiO_2_ (ratio)	60 (50–80)	100 (55–100)	1.000
Hb (g/dL)	9.8 (8.8–10.5)	9.6 (8.9–11.4)	0.764
Leucocytes (/nL)	13 (10–17)	11 (9–19)	0.558
Platelet (G/L)	222 (165–395)	131 (87–214)	0.074
pfHb (mg/L)	36 (22–50)	48 (30–72)	0.162
aPTT (s)	30 (29–45)	32 (29–53)	0.174
INR (ratio)	1.3 (1.2–1.4)	1.1 (1–1.2)	0.899
ATIII (%)	70 (57–78)	58 (40–75)	0.155
D-dimer (µg/dL)	4,345 (1810–11,806)	2,400 (1782–7030)	0.253
Fibrinogen (mg/dL)	493 (454–704)	364 (276–490)	0.000 *
PCT (percentage)	4 (2–6)	1 (0–6)	0.053
LDH (U/L)	403 (312–578)	466 (318–644)	0.589
ALT (U/L)	33 (29–44)	35 (22–50)	0.713
AST (U/L)	66 (38–140)	62 (32–164)	0.967
BUN (mg/dL)	66 (44–114)	61 (37–92)	0.395
Creatinine (mg/dL)	1.2 (0.9–1.7)	1.1 (0.7–1.8)	0.540
Bilirubin (mg/dL)	0.6 (0.45–1.9)	0.72 (0.51–1.66)	0.765
Respiratory Extracorporeal Membrane Oxygenation Survival Prediction Score			
Total RESP Score	0 (−3–2)	1 (−2–2)	0.096
Survival prediction (percentage)	57 (33–57)	57 (33–57)	0.131
Cardiac arrest before ECMO, *n* (%)	0 (0)	5 (15)	0.063
PaCO_2_ ≥ 75 mmHg, *n* (%)	6 (32)	14 (41)	0.424
Peak inspiratory pressure ≥ 42 cmH_2_O, *n* (%)	0 (0)	0 (0)	1.000
Non-infectious indication for ECMO, *n* (%)	0 (0)	1 (3)	1.000

Continuous variables are presented as the median and interquartile range (IQR). * *p* values < 0.05 are significant and tagged with asterisks. † Including all patients with an MDRD-GFR < 60 mL/min. Abbreviations: ALT—alanine transaminase; AST—aspartate transaminase; aPTT—partial thromboplastin time; BMI—body mass index kg/m^2^; BUN—blood urea nitrogen; COPD—chronic obstructive pulmonary disease; ECMO—extracorporeal membrane oxygenation; pfHb—plasma-free hemoglobin; Hb—Hemoglobin; ICU—intensive care unit; IJV—internal jugular vein; IL-6—interleukin-6; Jug—jugular vein; LDH—lactate dehydrogenase; LOS—length of stay; NCOVID—non-COVID; PCT—procalcitonin; WBC—white blood cells.

**Table 2 jcm-10-02547-t002:** Outcomes and clinical course.

	COVID (*n =* 19)	NCOVID (*n =* 34)	*p* Value
90-day mortality, *n* (%) †	9 (47)	25 (74)	0.064
ECMO duration (d)	16 (11–23)	13 (5–25)	0.714
Total LOS in-hospital (d)	29 (14–55)	33 (11–60)	0.576
Thromboembolic events, *n* (%)	8 (42)	4 (12)	0.031 *
Pulmonary artery embolism	5 (26)	0 (0)	0.008 *
Peripheral venous thrombosis	3 (16)	3 (9)	0.508
Peripheral arterial thrombosis	1 (5)	0 (0)	0.500
Other thromboembolic events ††	0 (0)	1 (3)	1.000
Major bleeding events, *n* (%)	8 (42)	21 (62)	0.263
Endobronchial	2 (11)	8 (24)	0.227
Mucosal	3 (16)	4 (12)	0.727
Cannulation side	2 (11)	9 (26)	0.267
Gastrointestinal	1 (5)	0 (0)	0.500
Cerebral	0 (0)	4 (12)	0.125
Hemothorax	1 (5)	4 (12)	0.625
Pericardial tamponade	2 (11)	0 (0)	0.125
Other adverse events, *n* (%)			
Acute kidney failure ^§^	13 (68)	17 (50)	0.093
Severe thrombocytopenia ^§§^	2 (6)	11 (32)	0.057
Hemorrhagic shock	0 (0)	4 (12)	0.125

Continuous variables are presented as the median and interquartile range (IQR). * *p* values under 0.05 are significant and tagged with asterisks. † Death during 90 days after initiation of ECMO therapy. †† One embolic liver damage was reported. ^§^ All patients with acute kidney failure in stages 2 or 3 by KDIGO guidelines. ^§§^ All patients who developed a platelet count under 50 during ECMO therapy. Abbreviations: ECMO—extracorporealembrane oxygenation; ICU—intensive care unit; LOS—length of stay, NCOVID—non-COVID.

## Data Availability

The data presented in this study are available on request from the corresponding author.

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
