# Peer review of "Outcomes of Extracorporeal Membrane Oxygenation for Acute Respiratory Distress Syndrome in COVID-19 Patients: A Propensity-Matched Analysis"

_jcm, 2021, doi:10.3390/jcm10122547_

Round 1

Reviewer 1 Report

In the present manuscript the authors describe the results of a propensity-matched analysis of extracorporeal membrane oxygenation treatment for acute respiratory distress syndrome in COVI-19 patients.

The topic is original and of significant and contemporary clinical importance and absolutely worth being published in the Journal of Clinical Medicine.

The Manuscript is extremely well written and the respective sections adequately chosen in lengths and content. The number of figures and tables is adequate.

There are only some remarks, which should be possible to be revised quickly.

  1. Even if the difference in outcome concerning 90-days mortality is not statistically significant, 47% vs 74% with p=0.064 may be described as a “trend” toward better survival within the COVID-19 group.
  2. Introduction and reference 5: “During the 2019 influenza A (H1A1) pandemic, ARDS patients benefited from ECMO support.” Since there was no H1A1 pandemic in 2019 and the manuscript mentioned is from 2011, the authors probably meant “2009”!?
  3. Material and Methods: “We adapted our anticoagulation strategy and aimed at higher ACT and aPTT targets” Please give target-values.
  4. Figure 1: This is more to the editor: the figure-legends are to small and the color differences to blurred to be judged by the reviewer.
  5. The authors describe more thromboembolic events (TEE) within the COVID-19 group as compared to the control-group. How were these TEEs diagnosed? Since the authors mentioned that autopsies of COVID-19 pats identified high incidences of TEE, were there also TEEs identified in survivors?
  6. Figure 2: There is no legend indicating which line belongs to which group. Therefore it may appear even more confusing that the upper line belongs to the (lower mentioned) COVID-group.

Author Response

1. Even if the difference in outcome concerning 90-days mortality is not statistically significant, 47% vs 74% with p=0.064 may be described as a “trend” toward better survival within the COVID-19 group.

Dear Reviewer, thank you for this advice. We agree with Reviewer and we added this information to the discussion.

Changes to the manuscript Discussion Page 9, line 282

On the contrary, we found a trend towards better survival in COVID-19 patients.

2. Introduction and reference 5: “During the 2019 influenza A (H1A1) pandemic, ARDS patients benefited from ECMO support.” Since there was no H1A1 pandemic in 2019 and the manuscript mentioned is from 2011, the authors probably meant “2009”!?

A2: Dear Reviewer, thank you for this valuable comment. Indeed, this was a typo. We changed it to 2009 in the Introduction (line 47)

3. Material and Methods: “We adapted our anticoagulation strategy and aimed at higher ACT and aPTT targets” Please give target-values.

A3: Dear Reviewer, thank you for this important hint. We meant to say that we used the same target values, but tried to reach the upper limits. We improved this sentence in line 95.

Changes in the manuscript Material and Methods page 3 lines 95-96:

We adapted our anticoagulation strategy and aimed at higher ACT and aPTT targets (PTT 50-60s and ACT 180s), this was also recommended by ELSO guideline

4. Figure 1: This is more to the editor: the figure-legends are too small and the color differences to blurred to be judged by the reviewer.

A4: We hope that the final version will be more appropriate, we did provide a high-resolution Figures separately, but we were also asked to add figures immediately to the review File

5. The authors describe more thromboembolic events (TEE) within the COVID-19 group as compared to the control-group. How were these TEEs diagnosed? Since the authors mentioned that autopsies of COVID-19 pats identified high incidences of TEE, were. there also TEEs identified in survivors?

A5: Dear Reviewer, thank you for this important comment. It seems, that we did not describe this point appropriately. With Autopsy we did refer to previous studies.

We made re-phrased the sentence.

       Changes in the manuscript

Material and Methods page 3, line 94:

As many recent studies have shown, that COVID-19 patients have an increased risk for thromboembolic events  [1-3].

Page 3, lines 114-116

When TEE were suspected based on clinical symptoms and physical examination,

 CT-scans and ultrasound were conducted to secure the diagnosis of TEE.

Results page 7, lines 287-288:

Approximately 50% of our patients with thrombotic events were survivors.

Discussion page 9, lines 338-339:

Autopsies of COVID-19 patients in recent studies identified high incidences of TEE [4, 5].

6. Figure 2: There is no legend indicating which line belongs to which group. Therefore, it may appear even more confusing that the upper line belongs to the (lower mentioned) COVID-group.

A6: Indeed, it is confusing. We added the information in the figure legend.

Reviewer 2 Report

The authors used a propensity-matched analysis to compare outcome of ECMO treatment in covid-19 with non-covid ARDS patients and found no differences in outcomes apart from more thromboembolic events in the covi-19 patients.

Major limitation is the very small sample size. Many publications with far more patients have been published since the outbreak of the covid-19 pandemic. Furthermore, severity of illness is very difficult to estimate in critically ill patients and bias by indication is huge.

Anticoagulation strategy differed between groups: in COVID-19 patients a higher ACT and aPTT was aimed. Which target and was it reached?

Was a CT-thorax performed in all non-covid ARDS patients to exclude PE?

Five (!) patients in the non-covid ARDS group had a cardiac arrest before ECMO, another 2 had leukemia. This is a very sick population with subsequent mortality of 74%.

Author Response

The authors used a propensity-matched analysis to compare outcome of ECMO treatment in covid-19 with non-covid ARDS patients and found no differences in outcomes apart from more thromboembolic events in the covid-19 patients.

Major limitation is the very small sample size. Many publications with far more patients have been published since the outbreak of the covid-19 pandemic. Furthermore, severity of illness is very difficult to estimate in critically ill patients and bias by indication is huge.

  1. Anticoagulation strategy differed between groups: in COVID-19 patients a higher ACT and aPTT was aimed. Which target and was it reached?

A 1: Dear Reviewer, thank you for this important comment. At the beginning of our ECMO-COVID series, we used the same target limits in both groups (PTT 40-50s and ACT 160-180s). However, due to the ELSO recommendation we aimed higher levels (upper limit of the targets) within the targets. However, in Figure 1, we reported that aPTT levels were significantly higher in the non-COVID-19 group 24 hours after ECMO-initiation (when compared as a single time-point). This is only one single time point. We do have the daily mean values of PTT from both groups.

We now present the time-course of PTT of both groups during the first week on ECMO as a supplementary Figure S2 and there is no significant difference between the two group during the treatment, as compared using two-way ANOVA with adjusted p-values for repeated comparison.

Changes to the manuscript: Supplementary Figure S2 has been added.

  1. Was a CT-thorax performed in all non-covid ARDS patients to exclude PE?

A2: Yes, all ARDS patients in our institution receive a CT-Scan as soon as possible. Some of the patients were stable prior to ECMO, other were first transportable after ECMO. So, all ARDS non-COVID have a CT-scan but not necessary prior to ECMO.

If there was a susception of PE, transthoracic and Transesophageal Echocardiography were immediately performed bedside, and if necessary emergent CT was performed as well

Changes in the manuscript:

Material and Methods page 3, lines 140-141-132:

If TEE were suspected based on clinical symptoms and physical examination, ultrasound, CT scans, and if necessary, MRI was conducted to secure the diagnosis of TEE.

  1. Five (!) patients in the non-covid ARDS group had a cardiac arrest before ECMO, another 2 had leukemia. This is a very sick population with subsequent mortality of 74%.

A 3: Dear Reviewer, we appreciate this important hint. Indeed, the cohort include critically ill patients, but as well as the COVID-group do.

We performed the propensity matching according to few risk factors, and we were not able to account for all risk factors, due to the small cohort.

Despite accounting for bias with propensity matching, the small number of patients leading to bias between the groups remain a limitation of or study.

We added this limitation to the discussion section.

Changes in the manuscript:

Discussion – Limitations page 10, lines 411-414: Despite accounting for baseline bias with propensity matching, we could not use all known risk factors in our propensity scores, therefore some degree of bias in patients’ characteristics remain evident between the two groups.

Reviewer 3 Report

Dear Editor,

thank you for the opportunity to review the manuscript “Outcomes of extracorporeal membrane oxygenation for acute respiratory distress syndrome in COVID-19 patients: A propensity-matched analysis” by Teresa Autschbach et al. submitted to Journal of Clinical Medicine. The paper is quite clearly written but several concerns remain:

  1. Major - The primary outcome of the study should be only one. You stated “Our primary endpoints were in-hospital mortality and adverse events occurring within 90 days after ECMO initiation”. Please consider which is your primary outcome.

  2. Major - The way you constructed the model of propensity match is questionable. You did not write why you chose those particular variables (arbitrary? P value <0.10?). (RESP score for instance is not recommended to predict mortality - cit. https://doi.org/10.3390/membranes11030170) At the end, in your model you have patients that are frankly not comparable (pH, FiO2, pCO2 are definitely different). Building a Kaplan Meier with these premises is questionable. Referring to your Table 1, you should correct for others confounders with a Cox regression.I suggest you to rebuild another model for propensity match with a better control of confounders.
  1. Major - If the outcome change (adverse events instead of mortality), you should rebuild another propensity score match table.

  2. Major - It is not clear why in your whole cohort of patients (prior to match) in COVID-19 group you have 18 patients in VV-ECMO and you considered 19 patients in the analysis. Concomitantly, in NCOVID group you have 105 patients and you have only 38 patients with VV-ECMO. (Supplemental Table 1). You stated in methods “We reviewed our institution database for all ARDS patients (age >18 years old) who received 65 VV-ECMO therapy between January 2015 and January 2020”. Please explain it.

  3. Minor - In univariate analysis, if you compare categorical variables not paired you should use Chi Square / Fisher’s exact test, as Mc Nemar is usually for paired categorical variables.

  4. Minor - In methods, even if you put reference you should explain, at least the configuration that you used for ECMO (F-F; F-J, bilumen)

Author Response

 1. Major - The primary outcome of the study should be only one. You stated “Our primary endpoints were in-hospital mortality and adverse events occurring within 90 days after ECMO initiation”. Please consider which is your primary outcome.

A 1: Dear Reviewer, thank you for your in-depth review and for this very important comment, we agree with reviewer, we cannot state that there are more than on primary end point. As we carried out the propensity matching the test of balance and goodness of the propensity model was based on one treatment (the groups) and covariates and one outcome, in our case it was the in-hospital mortality).

Dear Reviewer, thank you again for this valuable hint.

We made the appropriate changes to the manuscript.

Changes in the manuscript:

Material and Method:

Page 3, Lines 136-137

 Our primary endpoint was in-hospital mortality and the secondary end-points were major adverse events occurring within 90 days after ECMO initiation.

2. Major - The way you constructed the model of propensity match is questionable. You did not write why you chose those particular variables (arbitrary? P value <0.10?). (RESP score for instance is not recommended to predict mortality - cit. https://doi.org/10.3390/membranes11030170) At the end, in your model you have patients that are frankly not comparable (pH, FiO2, pCO2 are definitely different). Building a Kaplan Meier with these premises is questionable. Referring to your Table 1, you should correct for others confounders with a Cox regression.I suggest you to rebuild another model for propensity match with a better control of confounders.

A 2: Dear Reviewer, thank you for this valuable comment. We agree with reviewer, despite our propensity matching based on age, RESP Score, creatinine and bilirubin we were not able to account for all possible confounder, this is a result of the small COVID-19 cohort. We tried to account for more covariates in the propensity score, but this did lead to a loss of many COVID-19 patients, making the analysis inappropriate.

This is an important limitation, which we did mention now in the limitation section.

Beside gender and age, we did choose few baseline characteristics to reflect the organ function prior to ECMO, we chose creatinine as a measure of the kidney function, the bilirubin as a measure for liver function. We appreciate the suggestion from reviewer about the prediction value of RESP score in COVID-19 patients, in contrary to the provided Reference from reviewer, other groups found RESP Score to be a good predictor of survival of COVID-19 treated with ECMO [a) Yang X et al. Extracorporeal Membrane Oxygenation for Coronavirus Disease 2019-Induced Acute Respiratory Distress Syndrome: A Multicenter Descriptive Study. Crit Care Med. 2020;48(9):1289-1295

  1. Yang et al. “Extracorporeal Membrane Oxygenation in Coronavirus Disease 2019-associated Acute Respiratory Distress Syndrome: An Initial US Experience at a High-volume Centre.” Cardiac failure review vol. 6 e17. 26 Jun. 2020, doi:10.15420/cfr.2020.1
  2. C) Zayat, Rashad et al. “Role of extracorporeal membrane oxygenation in critically Ill COVID-19 patients and predictors of mortality.” Artificial organs, 10.1111/aor.13873. 24 Nov. 2020, doi:10.1111/aor.13873]

Dear Reviewer, we now do provide a figure showing selected baseline characteristics with the standardized Absolut bias difference before and after matching (Supplementary Figure S1)

With this figure we demonstrate the accepted balance between the covariates which were chosen for the matching.

We agree with reviewer, it was not appropriate to perform a simple Kaplan-Meier-analysis. We now present the result of Kaplan-Meier survival analysis weighted for the propensity score and to test the difference between the groups an adjusted Fleming-Harrington test instead of log-rank. In addition, we performed parametric survival regression adjusted to the propensity scores to estimate the Hazard ratio.

Changes in the manuscript:

Material and Methods:

Statistical analysis page 3 lines 152-204:

From 20 ARDS COVID-19 patients one patients had VA-ECMO and were excluded from the analysis. In the non-COVID ARDS group 38 (36%) patients had VA-ECMO and were not included in matching. We used a 1:2 matching approach with a maximum score radius of 0.05 to match between 19 ARDS COVID-19 patients (COVID-group) and 67 non-COVID ARDS patients (NCOVID-group). Four patients from COVID-group were able to be matched with one patient from NCOVID-group. After matching we had 19 patients in the COVID-group and 34 patients in the NCOVID-group. The balance between the matched group was assessed with the standardized absolute bias difference (supplementary Fig. S1). This matching procedure enabled higher precision at little cost of bias. Comparisons between unmatched groups were performed with two-tailed Student’s t-tests for normally distributed continuous variables and with the Mann-U test for non-normally distributed continuous variables. Categorical variables were analysed with a chi-square test or, if appropriate, Fisher’s exact test. In the matched cohort, univariate analyses were conducted using the Wilcoxon signed-rank test for continuous variables and McNemar’s test for categorical variables. Analyses of laboratory parameters at three time points were conducted using Friedman’s nonparametric test with Dunn’s correction for repeated measurement. Adjusted p values for multiple corrections are presented. crude survival by COVID-19 status was assessed with Kaplan–Meier analyses weighted for the propensity scores and the test of equality of survival was carried out through Fleming-Harrington test parametric survival regression adjusted for the propensity score was performed estimate the hazard ratio (HR) and corresponding 95% confidence interval (CI). All statistical comparisons were two-sided, and a p value of <0.05 was significant.

Results page 7, lines 290-292:

Table 2 presents all outcomes and complications. During a 90-day follow-up, mortality was 47% in the COVID-group and 74% in the NCOVID-group (Fleming-Harrington test of equality of survival, p=0.153). Figure 2 presents the Kaplan–Meier survival curves weighted for propensity score. In the adjusted parametric survival regression patients in the COVID-group had a hazard ratio of 3.1, p=0.056 [95%-confidence interval: 0.99-9.96].

Limitations: page 10, lines 411-414

Despite accounting for baseline bias with propensity matching, we could not use all known risk factors in our propensity scores. Therefore, some degree of bias in patients’ characteristics remains evident between the two groups.

3. Major - If the outcome change (adverse events instead of mortality), you should rebuild another propensity score match table.

A3: dear reviewer, you have absolutely right, but as we stated in our Answer 3 to your question 3, the primary outcome was in-hospital mortality.

4. Major - It is not clear why in your whole cohort of patients (prior to match) in COVID-19 group you have 18 patients in VV-ECMO and you considered 19 patients in the analysis. Concomitantly, in NCOVID group you have 105 patients and you have only 38 patients with VV-ECMO. (Supplemental Table 1). You stated in methods “We reviewed our institution database for all ARDS patients (age >18 years old) who received 65 VV-ECMO therapy between January 2015 and January 2020”. Please explain it.

A4: Dear Reviewer, thank you for this important note. There was a mistake in presenting the information that 20 COVID-19 patients were available and one of them had VA-ECMO, the patients with VA-ECMO was excluded from the matching. In the NCOVID-group 38 patients had VA-ECMO and were excluded from the matching.

Supplementary Table S1 have been re-checked and corrected. The information about VA-ECMO patients have been added to the main text.

Changes in the manuscript:

Statistical analysis:

Page 3 lines 152-154

From 20 ARDS COVID-19 patients one patients had VA-ECMO and were excluded from the analysis. In the non-COVID ARDS group 38 (36%) patients had VA-ECMO and were not included in matching.

Supplementary Table S1 has been re-checked and corrected

5. Minor - In univariate analysis, if you compare categorical variables not paired you should use Chi Square / Fisher’s exact test, as Mc Nemar is usually for paired categorical variables.

A5: Dear Reviewer, we used Fisher’s exact test for the unmatched analysis and Mc Nemar for the matched group. We added this information to the main text:

Changes in the manuscript:

Page 3, lines 161-164

Comparisons between unmatched groups were performed with two-tailed Student’s t-tests for normally distributed continuous variables and with the Mann-U test for non-normally distributed continuous variables. Categorical variables were analyzed with a chi-square test or, if appropriate, Fisher’s exact test

6. Minor - In methods, even if you put reference you should explain, at least the configuration that you used for ECMO (F-F; F-J, bilumen) 

A6: Dear reviewer, we added the requested information.

Changes to the manuscript: page 2, lines 88-96

Percutaneous cannulation with the Seldinger technique was our preferred technique for VV ECMO. Depending on the desired flow rate and where possible, bi-caval cannulation with a double lung cannula (27 to 31 Fr) was performed preferentially to two-site can-nulation (femoral-jugular or femoral-femoral) with 19 to 25 Fr cannulation. The decision whether to perform double-lumen cannulation on a single-site or two-place cannulation depends on many factors. In short, a 25 Fr vein cannula as a drainage cannula is generally necessary in a patient with high BSA (2.2-2.5 m2), and a 17-19 Fr vein cannula for venous return is appropriate to achieve sufficient flow with adequate carbon dioxide clearance and oxygenation.

Round 2

Reviewer 2 Report

none

Reviewer 3 Report

No further comments from my side, thank you for your extensive revision.